# Validation of Spanish Erasmus-Modified Nottingham Sensory Assessment Stereognosis Scale in Acquired Brain Damage

**DOI:** 10.3390/ijerph182312564

**Published:** 2021-11-29

**Authors:** Belén Dolores Zamarro-Rodríguez, Miguel Gómez-Martínez, César Cuesta-García

**Affiliations:** 1Occupational Thinks Research Group, Centro Superior de Estudios Universitarios La Salle, Universidad Autónoma de Madrid, 28023 Madrid, Spain; miguel.gomez@lasallecampus.es (M.G.-M.); cesar.cuesta@lasallecampus.es (C.C.-G.); 2Occupational Therapy Department, Centro Superior de Estudios Universitarios La Salle, Universidad Autónoma de Madrid, 28023 Madrid, Spain; 3Instituto de Rehabilitación Funcional La Salle, 28023 Madrid, Spain

**Keywords:** stereognosis, validation study, upper extremity, stroke, somatosensory disorders

## Abstract

Acquired brain injury (ABI) is the third leading cause of death in Spain. The disability derived from ABI can include considerable difficulties in upper-limb use due to somatosensory deficits. One of the assessments most commonly used to evaluate ABI is the Nottingham Sensory Assessment (NSA); however, there is no complete psychometric analysis or standardized version in Spanish. We aimed to develop and validate a Spanish version of the stereognosis component of the NSA for evaluating Spanish adults with ABI via a single-center, observational, cross-sectional study. The Spanish version of the NSA was developed in two steps. The first was based on the standardization and collection of normative data in 120 asymptomatic participants. For the second, we recruited 25 participants with ABI to establish concurrent criterion-related validity, internal consistency, and floor/ceiling effects. Criterion validity was assessed against two-point discrimination and tactile-localization tests. Our normative data showed significant differences among the various age groups (*p* < 0.05), supporting the validity of the Spanish-version assessment. For the ABI sample, we also found further evidence of validity with Spearman’s rho coefficient between the total scores and the two-point discrimination and tactile-localization tests, which showed low and moderate correlations (rho = 0.50–0.75, *p* < 0.05). Internal consistency was excellent, with a Cronbach’s alpha of 0.91. No ceiling or floor effects were found. We conclude that the stereognosis component of the NSA in its Spanish version is a valid scale that can be used to comprehensively and accurately assess stereognosis capacity in adults with ABI. As a low-cost evaluation, this assessment has great potential to be widely used in clinical practice and research settings.

## 1. Introduction 

Acquired brain injury (ABI) is an isolated disease determined, in 78% of cases, by either external causes (such as cranioencephalic trauma) or internal causes, among which strokes are the most frequent [1]. When a brain injury occurs, it is accompanied by a series of limitations, which can be disabling. These limitations include a variety of motor, perceptual, sensory, language, psychological, and behavioral deficits [2].

Spain currently has an annual ABI incidence of 187 cases per 100,000 inhabitants, with a higher incidence in women than in men [3]. Not only is ABI the third leading cause of death in Spain, but most cases involve significant disability, dependence, and the need for hospital stays. The disability presented represents up to 60% of difficulties in the use of the affected upper limb (UL) [4,5,6]. These difficulties in the UL can be multiple and varied, with 55% of the population having somatosensory deficits from the onset of the disease [7].

Somatosensory information allows individuals to adapt to tasks and to the environment. Its affectation implies a greater alteration in the motor function of the UL; it lasts beyond the acute phase and translates into a decrease in the spontaneous use of the affected limb. Given that this difficulty usually continues for several months, repercussions on an individual’s daily performance and quality of life are inevitable [8]. 

Among the sensory deficits, ABI in most (63%) cases is the cause of major impairments in combined cortical sensitivity, mainly stereognosis ability and two-point discrimination [9]. 

Although clinicians and researchers recognize that the assessment of somatosensory processing is essential to the prognosis of functional capacity for patients with ABI, diagnostic tests with a high degree of validity are scarce. Overall, the scientific literature lacks complete psychometric analyses for such assessments [10].

In clinical practice, one of the most widely used batteries is the Nottingham Sensory Assessment (NSA), created by Lincoln et al. in England in 1991. The NSA was developed to identify sensory deficits as well as to monitor the recovery process in cerebrovascular diseases [11]. In 2006, Erasmus MC modified the tool, known as the Erasmus MC modified Nottingham Sensory Assessment (EmNSA). This modified version incorporated the stereognosis subscale and improved the standardization and reproducibility in most of the assessments, providing better psychometric criteria, especially in terms of reliability [12]. However, there is still a scarcity of data regarding its complete analysis, with a lack of normative data, reliability processes, validation, and cohort scores.

In 2012, Connell et. al, analyzing the available sensitivity tools, concluded that the NSA was one of the best somatosensory tools available, together with the Fugl-Meyer, due to its easy administration, low cost, and portability. However, the current tendency to use superficial assessments with poorly structured protocols and questionable reproducibility could be problematic. For this reason, in this project, we aimed to culturally adapt and validate the stereognosis subscale of the NSA tool in a Spanish population diagnosed with ABI.

## 2. Materials and Methods

### 2.1. Cross-Cultural Adaptation 

We aimed in this first phase to adapt and standardize the administration of the NSA subscale, providing evidence of validity in the Spanish population. Following the guidelines of the International Test Commission [13], the following steps were taken:

#### 2.1.1. Authorization of the Original Author

Once the study was approved by the Ethics Committee of the Centro Superior de Estudios Universitarios La Salle (Madrid), authorization was obtained from the Nottingham Assessment company and the authors for adaptation of the tool in a Spanish version.

#### 2.1.2. Adaptation and Standardization of the Material Used

Following the recommendations established by the authors and the recent data provided in the scientific literature, the administration of the test was protocolized through the use of a frame, adding a variable capturing total time employed on the task and establishing an exchange of the currencies used for their equivalent value in EUR.

#### 2.1.3. Pilot Study

We began by administering the test to 30 participants of Spanish nationality without pathology to determine possible errors during its administration, ultimately expanding the sample to 120 participants from whom we collected normative data. Participants were also asked about the relevant sensory qualities when recognizing each object.

### 2.2. Validation Phase

A cross-sectional observational study was conducted to determine the psychometric properties of the translated instrument. The project was developed at the Centro de Referencia Estatal de Atención al Daño Cerebral (CEADAC) of the Community of Madrid from March 2018 to August 2021. The ethics committee of the Centro Superior de Estudios Universitarios La Salle (No. Ref. CSEULS-PI-031/2020) approved the study. All participants provided informed consent for data collection. 

### 2.3. Participants

The participants were individuals with hemiparesis secondary to ABI, selected by the center’s rehabilitation physician, following the inclusion and exclusion criteria established by the principal investigators. All those aged 18 years and older and diagnosed with ABI were included if they presented a cognitive level with scores above 24 on the Mini-Examen Cognoscitivo cognitive assessment; had muscle tone with mild or moderate hypertonia; scored 1, 1+, or 2 on the modified Ashworth scale; and had sufficient motor control in the affected hand, being able, once seated, to pick up a cloth and make the movement of cleaning the table and pick up a small object with any type of tweezers. Participants were excluded who were not clinically stable; who had behavioral problems; who had brain damage of neurodegenerative origin; who had hemineglect and/or mixed aphasia; or who had peripheral nerve lesions of the upper extremity.

### 2.4. Procedure

The assessment process began with an interview to collect sociodemographic data of interest (sex, age, type of injury, time of injury, dominant hand, and upper limb most affected). To perform the various sensory tests, a frame was used (see Figure 1). The frame made it possible to place each participant in a seated position close to a table and to avoid having to use any mask that would generate additional sensory changes, which could cause discomfort.

The frame used consisted of a 61 × 40 cm ethylene-vinyl acetate (EVA) rubber surface, where the hand to be evaluated was located, eliminating any auditory feedback that might be generated when depositing the evaluation material. This EVA rubber surface included two 40 × 7 × 2 cm wooden strips at 20 cm from the edge of the table, joined by a dark cloth to occlude the vision. For the stereognosis subtest, we placed two panels on both sides of the slats with a surface divided into 11 sections where the various objects were placed. 

### 2.5. Measuring Instruments 

For this project, the stereognosis and tactile-localization subscales of the EmNSA as well as a two-point discrimination test were administered. 

The EmNSA stereognosis subscale assesses the haptic recognition ability of 11 everyday objects through the affective UL. The scoring system is determined by a range of scores from 0 to 2, in which 0 points are given for the inability to recognize the object (stereognosis), 1 point when the participant can determine some of the characteristics of the object, and 2 points for correct recognition. Thus, it is possible to achieve a total score between 0 and 22 points. The total time spent on the recognition of the objects and the exchange of coins for their value in EUR were added for the Spanish version. 

**Esthesiometer**, two-point discrimination test was performed using a BASELINE manual esthesiometer on the thumbs of the first, second, and fifth fingers, as well as the thenar and hypothenar areas of the affected UL. The minimum distance at which the participant distinguished the two stimuli was determined by pressing simultaneously for approximately 0.5 s. The procedure was performed in decreasing intervals.

The EmNSA tactile-localization subscale. With a fine-tipped pen, pressure was applied to the same regions of the hand as in the previous test, with the aim that, once deprived of visual input, the participant would indicate where exactly they felt the stimulus to measure the centimeters of distance in their response, using a manual esthesiometer.

### 2.6. Statistical Analysis 

The data analysis was performed with SPSS version 22.0 software (IBM SPSS Statistics for Windows, IBM, Armonk, NY, USA). In the cross-cultural adaptation phase, all variables were assumed to be normally distributed following the central limit theorem, given that all groups had at least 30 participants. Descriptive statistics were used to summarize the data for the continuous variables, which are presented as mean ± standard deviation and 95% confidence interval, and *p* < 0.05 was considered significant. A one-factor analysis of variance was used to compare the means of the control variables between groups in the asymptomatic sample, and we also performed a between-group comparison. 

In the second part of this study, a sample size of 25 patients was determined, in which an effect of 0.6 with a power of 90% was used to explain the observed correlation between the variables. For the concurrent validation analysis, correlations of the independent variables were performed with Spearman’s rho tests, whose scores were classified as low (r < 0.50), moderate (r = 0.50–0.75), or high (r > 0.75) [14]. Cronbach’s α coefficient was calculated with values interpreted as excellent (>0.8), acceptable (0.7–0.8), or low (<0.7) [15]. Finally, the floor and ceiling effect was considered significant if more than 20% of the sample achieved scores of 0 or 22 [16].

## 3. Results

### 3.1. Cross-Cultural Adaptation 

A total of 120 participants were included in our pilot study of normative-data collection, in which significant differences (*p* < 0.05) were obtained among the various age groups for the total score obtained in the stereognosis subtest (Table 1).

In the first statistical analysis, there were no differences found in terms of dominance per hand for the different groups; therefore, although right-handed participants were predominant, the data were grouped independently of the participant’s dominant hand. 

Although we observed similar total scores, more mistakes were made as the age of participant increased. The highest number of mistakes was observed for the elderly group, reaching a minimum total value of 14 points, which implies failures for a minimum of eight objects. These inter-group differences for the total scores were statistically significant (*p* = 0.02) (Table 2). 

The greatest difficulties were observed for recognition of the three coins. All the groups presented high percentages of mistakes for the recognition of all the coins, although the greatest difficulty was in distinguishing the EUR 0.5 coin, which, for the groups of young people (43.3%), adults (60%), middle-aged (63.3%), and seniors (70%), was easily confused with the EUR 2 coin. The group of older adults also encountered great difficulties with the distinction of this coin (56.7%).

Another combination of objects to be highlighted is the glass and the cup, for which their distinction was also challenging. Difficulties were observed both in the middle-aged group, with 6.7% lacking the ability, as well as 10% in the group of adults. 

Similarly, we found that, in the middle-aged and seniors groups, those objects that contained a greater load of sensory information also posed a difficulty when it came to recognition for 3.3% of the sample studied.

### 3.2. Validation Phase

The characteristics and clinical scores of 25 participants are shown in Table 3.

### 3.3. Concurrent-Criterion Validity 

A moderate negative and significant correlation was obtained between the total score obtained in the stereognosis subscale and the tactile localization of the thumb. 

On the other hand, significant positive low-to-moderate correlations were found for the relationship between the total time spent in the stereognosis subscale and tactile localization in the thumbs of the index and little finger, thenar area, and two-point discrimination in the thumb of the index finger. Scores ranged from 0.029 to −0.689, as shown in Table 4.

### 3.4. Internal Consistency, Floor and Ceiling Effect 

We estimated a Cronbach’s alpha coefficient of 0.91 (95% CI), suggesting excellent internal consistency for the total stereognosis subscale. Modifying any object of the study, Cronbach’s alpha varied very little; for example, when eliminating the scissors item, the increase was only 0.007 points. The item–total correlation coefficients ranged from 0.36 to 0.90, with the scissors item having the lowest correlation and the comb item the highest.

No ceiling or floor effects were observed. None of the participants reached the minimum (0 points), and the maximum (22 points) was reached by only 8% of the sample. The sample range was between 4 and 22 points.

## 4. Discussion

The preliminary validity of the stereognosis subscale in the Spanish context is promising and helps to solve one of the major shortcomings of the EmNSA. Contributions made by Connell and Tyson [17] in 2011 raised concerns that the tool still presents great psychometric deficiency in terms of translation, adaptation, and validation, highlighting the lack of normative-data collection, which gives more weight to the results obtained.

Regarding normative-data collection, our findings suggest a decrease in the total scores of the stereognosis subscale, finding differences between the various age groups. Exteroceptive sensitivity is essential to achieve good manual function [18]; as time passes, however, these abilities deteriorate, which caused us to find a wide variability in our results for the older age group [19,20].

With increasing age, participants have reported fewer key sensory qualities for the recognition of various objects. Texture and concrete shape are the qualities used by most of them, including in some groups the use of hardness, weight, or volume. For example, in the case of coins, texture (40.8%), volume (47.5%), weight (40.8%), and shape (73.3%) are essential for their recognition. On the other hand, if there is one thing that characterizes both the glass and the cup, it is that they have a specific shape (80% and 89.2%) that differentiates them. This lack of sensory analysis reflects their haptic capacity. 

Although there are few studies reported with the Spanish population, in 2020, Peña González et al. [21] provided data on the relationship between cognitive capacity, stereognosis, and manual dexterity in older people, again achieving poorer results with increasing age. These results were influenced by impairments in implicit cognitive and motor abilities during haptic object recognition. 

Although many of the objects explored do not require such specific cognitive or perceptual abilities, difficulties were observed in all age groups in the distinction of different coins. These results were also found by M. Reidy et al. in 2016 [22], who, when studying the haptic recognition of objects in asymptomatic individuals, also found greater difficulties regarding the identification of the EUR 0.5 and EUR 2 coins. Similarly, as the age of the participants increased, the distinction of these coins proved to be a more complex process, producing a higher percentage of failures in the hand with poorer manipulative dexterity.

If we analyze in detail the process of recognizing a coin, we observe that it is necessary to start with a closing/pressing movement, with which we first determine its size and overall shape, making the hand mold to the object and quickly providing spatial information. This is followed by lateral movements accompanied by contour tracking to acquire the most information about specific details of the object. A combination of movements is necessary to obtain information on the texture, volume, and shape of the object.

However, as Lederman and Klatzky [23] determined in 1987 during haptic object recognition, a series of common exploratory motor patterns are produced, which are influenced by the sensory qualities of each of the objects, given that two haptic subsystems come into play during the recognition process of three-dimensional objects: a sensory subsystem with cutaneous, thermal, and kinesthetic receptors; and a motor subsystem whose main function is to actively manipulate the objects. If the hand can adequately use its motor capabilities to facilitate its perceptual and cognitive functions, the movement patterns that will be performed will depend on the type of haptic information that has been processed. 

For our study sample, the sensory qualities analyzed became fewer with increasing age, so that the exploratory pattern was more limited, thus hindering the recognition process.

Similar data, in the stereognosis capacity, can be found for the ABI sample, in which 60% of the cases had difficulties in the processing of haptic information [24].

These findings make it important to consider the sensory qualities of the various objects to be explored when our goal is to trigger a particular exploratory movement. 

On the other hand, regarding the results obtained in the validation phase, we found low to moderate associations for the sensitivity variables studied in the concurrent criterion validation, which reached statistically significant values.

In 2016, Cuesta-García C. [25] determined the tactile discrimination threshold to pressure in fingers I and V. Stereognosis and two-point discrimination in finger I and the hypothenar area were sensory variable predictors (R^2^ = 0.620; *p* < 0.01) of manual dexterity in patients with ABI in Spain. Our results agree with those findings, reporting moderate associations between the difficulties present in two-point discrimination and tactile localization of those hand regions, causing an impact on the efficiency and total time spent in haptic exploration. 

Like Carey L. in 2011, we can affirm that the sensory loss present in the studied sample causes an impact on their immediate exploration ability [26].

Although the literature reflects few studies in which complete psychometric analysis of the subscale has been performed, the present study shows excellent internal consistency through the Cronbach’s alpha coefficient, with data similar to the study conducted by Villepinte C. et al. [27] in 2018, which, after its adaptation to the French culture, obtained a score of 0.94 for its internal consistency. However, this is not the only study in which good results were obtained; in 2010, the adaptation to Brazilian cross-culture took place [28], and moderate-to-high results were also shown for its internal consistency in patients with acute-phase stroke.

Lastly, the results of the present study demonstrate how the use of a frame facilitates and standardizes the process of assessing stereognostic capacity. It provides more space for the development of exploratory motor patterns, allowing the assessment of hands in any physical condition.

Although such standardization reports promise validation data in a low-cost, rapid, and easy-to-administer tool, we found several limitations, such as measurement biases due to the lack of sensitivity of the instrument, which has limited the completion of its psychometric analysis. The tendency to assess stereognosis in a rudimentary manner means that we have not found any gold-standard instrument with which to complete the validation process of the subscale. On the other hand, difficulties were encountered during the recruitment process due to the current COVID-19 pandemic. 

However, these findings open the door to a contribution of clinimetric studies that will increase the clinical usefulness of the subscale. 

## 5. Conclusions

The results of this first study show promising data to improve the quality of assessments and subsequent clinical practice of health professionals, allowing us to broaden the field of tools for the assessment of stereognostic capacity, thus improving care for patients with ABI.

The Spanish version of the EmNSA stereognosis subscale shows good concurrent validity and internal consistency in the assessment of people with ABI. For the first time, it reports normative data and includes the performance-time variable in both asymptomatic persons and in those with neurological pathology.

This study’s conclusions support the use of the Spanish version of the stereognosis subscale (EmNSA) in daily clinical practice, because it is a low-cost tool based on scientific evidence.

## Figures and Tables

**Figure 1 ijerph-18-12564-f001:**
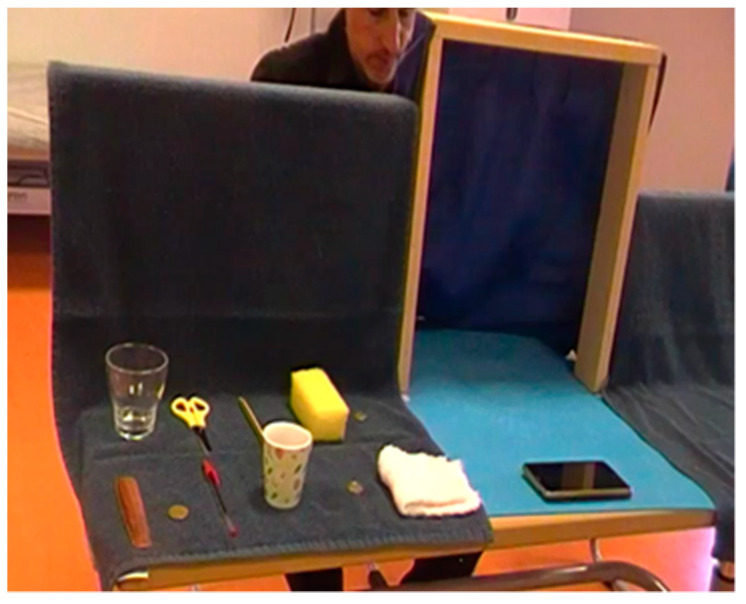
Frame and standardization of the stereognosis subscale.

**Table 1 ijerph-18-12564-t001:** Sociodemographic characteristics and stereognosis subtest scores for validation in the Spanish population (*n* = 120).

	Youth (18–30 Years) *n* = 30	Adults (31–49 Years) *n* = 30	Middle-Aged (50–65 Years) *n* = 30	Seniors (>65 Years) *n* = 30	*p*-Value
Age mean	22.73 ± 2.8 sd	44.43 ± 9.4 sd	56.77 ± 4.3 sd	78.30 ± 6.3 sd	
Sex (*n*%)					
Male	12 (60%)	15 (50%)	18 (40%)	21 (70%)	
Female	18 (40%)	15 (50%)	12 (60%)	9 (30%)	
Dominance (*n*%)					
Right-handed	26 (86.7%)	28 (93.3%)	16 (53.3%)	30 (100%)	
Left-handed	4 (13.3%)	2 (6.7%)	14 (46.7%)		
Hand rated (*n*%)					
Dominant	18 (60%)	23 (76.7%)	14 (46.7%)	14 (46.7%)	
Non-dominant	12 (40%)	7 (23.2%)	16 (53.3%)	16 (53.3%)	
TS	20.87 ± 0.9 sd	19.90 ± 1.2 sd	20.23 ± 1.1 sd	19.67 ± 1.1 sd	
Value Min–Max	19–22	17–22	18–22	14–22	0.02
TT	82.68 ± 26.1 sd	75.78 ± 21.0 sd	91.02 ± 27.8 sd	79.94 ± 37.1 sd	
Value Min–Max	39–136	47–126	43–160	26–161	0.21

TS: stereognosis subtest total score (NSA); TT: total time for stereognosis subtest (NSA); sd: standard deviation.

**Table 2 ijerph-18-12564-t002:** Mean of comparison between groups (mean ± standard deviation).

Comparison Groups	Variable	Difference in Means	95% Confidence Interval	*p*-Value
A. Youth vs. Adults	TS	0.96 *	0.14 to 1.80	0.01
B. Youth vs. Middle Aged	TS	0.63	−0.20 to 1.46	0.19
C. Youth vs. Seniors	TS	1.20 *	0.37 to 2.03	0.00
A. Youth vs. Adults	TT	6.89	−12.35 to 26.14	0.78
B. Youth vs. Middle Aged	TT	−8.34	−27.59 to 10.90	0.67
C. Youth vs. Seniors	TT	2.73	−16.51 to 21.98	0.98

* *p* < 0.05; TS: stereognosis subtest total score (NSA); TT: total time for stereognosis subtest (NSA).

**Table 3 ijerph-18-12564-t003:** Characteristics and clinical scores (*n* = 25).

Age mean	43.42 ± 9.61
Sex (*n*%)	
Male	17 (68%)
Female	8 (32%)
Type of ABI (*n*%)	
TBI	3 (12%)
Hemorrhagic stroke	10 (40%)
Ischemic stroke	9 (36%)
Tumor	3 (12%)
Dominance (*n*%)	
Right-handed	19 (76%)
Left-handed	6 (24%)
Hand rated (*n*%)	
Dominant	19 (76%)
Non-dominant	6 (24%)
EmNSA-SS/ST UL	
Discrimination two-point thumb (cm)	6.2 ± 5.0 1–20
Discrimination two-point index finger (cm)	9.5 ± 6.2 0.6–20
Discrimination two-point little finger (cm)	7.5 ± 5.3 0.5–20
Discrimination two-point thenar area (cm)	10.1 ± 6.3 1–20
Discrimination two-point hypothenar area (cm)	9.5 ± 6.2 0.6–20
Stereognosis subtest TS	18 ± 4.5 4–22
Stereognosis subtest TT (sec.)	206.4 ± 147.3 84–534
Tactile Location	
Tactile location thumb (cm)	0.5 ± 1.4 0–7
Tactile location index finger (cm)	1.2 ± 1.7 0–7
Tactile location little finger (cm)	0.4 ± 1.1 0–4
Tactile location thenar area (cm)	1.6 ± 2.3 0–8
Tactile location hypothenar area (cm)	1.2 ± 1.7 0–7

EmNSA-SS/ST UL: Erasmus-modified Nottingham Sensory Assessment Somatosensory/stereognosis components for the upper limb. Stereognosis subtest TS: stereognosis subtest total score (NSA); Stereognosis subtest TT: total time for stereognosis subtest (NSA); TBI: traumatic brain injury.

**Table 4 ijerph-18-12564-t004:** Concurrent criterion validation, Spearman’s correlation coefficient between stereognosis subtest, two-point discrimination, and tactile localization (*n* = 25).

EmNSA-SS	Tactile Location					Discrimination Two-Point				
	I	II	V	TA	HA	I	II	V	TA	HA
EUR 0.1 coin	−0.609 **	-	-	-	-	-	-	-	-	-
EUR 2 coin	-	-	0.427 *	0.544 **	-	-	-	0.427 *	0.544 **	-
Comb	-	−0.487 *	-	−0.550 **	-	-	-	-	-	-
Scissors	-	−0.414 *	−0.541**	−0.528 **	-	-	-	-	-	-
Pencil	−0.666 **	-	-	−0.613 **	-	-	-	-	-	-
Sponge	−0.472 *	−0.488 *	-	-	-	-	-	-	-	-
Glass	−0.493 *	−0.461 *	-	-	-	-	-	-	-	-
Cup		−0.490 *	-	−0.550 **	-	-	-	-	-	-
Stereognosis TS	−0.689 **	-	-	-	-	-	-	-	-	-
Stereognosis TT	-	0.029 *	0.411 *	0.627 **	-	-	0.167 **	-	-	-

EmNSA-SS: Erasmus-modified Nottingham Sensory Assessment Somatosensory component for the upper limb; I: thumb; II: index; V: Little finger; TA: thenar area; HA: hypothenar area; * *p* < 0.05; ** *p* < 0.01.

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
