# Peer review of "Validation of Spanish Erasmus-Modified Nottingham Sensory Assessment Stereognosis Scale in Acquired Brain Damage"

_ijerph, 2021, doi:10.3390/ijerph182312564_

Round 1

Reviewer 1 Report

Thank you for the opportunity to read this very interesting paper. This is a good paper with appropriate research procedures. In the future, I expect that you will conduct a longitudinal survey as well. I would like to suggest a minor comment for publication. Please revise it.

・Table

The number of significant digits should be the same.

Please correct "Confidence interval 95%" in Table 2 to "95% Confidence interval".Table 2 

Table 4 shows a mixture of "," and ". are mixed up.

How about making the 50-65 age group "middle aged"?

The table is difficult to read, so please adjust it.

・I think the submission guidelines instruct the session to describe the dataset availability. Please add it.

・State the limitations of your research in the discussion.

Author Response

Thank you for your review and comments, the proposed terminology and incorrect punctuation marks in the tables have been changed. Adjustments were also made to the tables to make them easier to read.

The last section of the discussion contains the limitations encountered in the development of the study.

Please find attached the modified article.

Reviewer 2 Report

Thank you for the opportunity to review this valuable work.

It is necessary to list other tools used for sensory assessment in the introduction section. Also, please explain to the reader why you chose the Nottingham Sensory Assessment (NSA).

The discussion section would be helpful if the author provided more information about the implications of the findings of the Spanish version of the NSA. Authors should also emphasize what readers can learn from their research.

Author Response

Thank you for your review and comments, a full review of the English language throughout the document was performed through a university collaborating agency, therefore, an extension was requested in the delivery of the corrections.

Another baseline assessment was included in the last paragraph of the introduction and the rationale for the choice of the NSA assessment as well as the further implication of the findings achieved at the clinical practice level.

I attach the modified article
